# Emerging Medical Treatment for Hypertrophic Cardiomyopathy

**DOI:** 10.3390/jcm10050951

**Published:** 2021-03-01

**Authors:** Alessia Argirò, Mattia Zampieri, Martina Berteotti, Alberto Marchi, Luigi Tassetti, Chiara Zocchi, Luisa Iannone, Beatrice Bacchi, Francesco Cappelli, Pierluigi Stefàno, Niccolò Marchionni, Iacopo Olivotto

**Affiliations:** 1Cardiomyopathy Unit, Careggi University Hospital, 50134 Florence, Italy; Mattiazampieri29@gmail.com (M.Z.); martina.berteotti@gmail.com (M.B.); alb.marchi@yahoo.com (A.M.); luigi.tassetti.1990@gmail.com (L.T.); zocchi.chiara@gmail.com (C.Z.); cappellifrancesco@inwind.it (F.C.); iacopo.olivotto@gmail.com (I.O.); 2Cardiac Surgery, Careggi University Hospital, 50134 Florence, Italy; liannone005@gmail.com (L.I.); beatricebacc@gmail.com (B.B.); pierluigi.stefano@unifi.it (P.S.); 3Department of Experimental and Clinical Medicine, University of Florence, 50121 Florence, Italy; niccolo.marchionni@unifi.it

**Keywords:** hypertrophic cardiomyopathy, left ventricular outflow tract obstruction, mavacamten, septal reduction therapy

## Abstract

Hypertrophic cardiomyopathy (HCM) is a common myocardial disease characterized by otherwise unexplained left ventricular hypertrophy. The main cause of disabling symptoms in patients with HCM is left ventricular outflow tract (LVOT) obstruction. This phenomenon is multifactorial, determined both by anatomical and functional abnormalities: myocardial hypercontractility is believed to represent one of its major determinants. The anatomical anomalies are targeted by surgical interventions, whereas attenuating hypercontractility is the objective of old and new drugs including the novel class of allosteric myosin inhibitors. This review summarizes the current treatment modalities and discusses the emerging therapeutical opportunities focusing on the recently developed cardiac myosin ATPase inhibitors Mavacamten and CK-274. Novel surgical and interventional approaches are also discussed.

## 1. Introduction

Hypertrophic cardiomyopathy (HCM) is a global genetic heart disease with 1:500 prevalence in the general population, defined by left ventricular hypertrophy in the absence of abnormal loading conditions [1]. The disease is characterized by wide heterogeneity at almost every level, from genetic basis to clinical expression. HCM has an autosomal dominant inheritance pattern and is associated with more than 2000 mutations in genes encoding sarcomere proteins, the most frequently involved being beta myosin heavy chain and myosin-binding protein C [2]. It shows incomplete penetrance and variable expressivity; the distribution of left ventricular (LV) hypertrophy preferentially affects the basal septum and anterior wall, although its morphological patterns and severity are extremely different involving different regions of the left and right ventricle, even within the same family [3]. The clinical course of HCM is characteristically diverse, ranging from completely asymptomatic subjects to advanced heart failure (HF) due to severe left ventricular outflow tract (LVOT) obstruction, restrictive physiology or left ventricular systolic dysfunction [4]. LVOT obstruction (LVOTO) is the main cause of exercise disability and an independent predictor of adverse events such as progressive HF and cardiovascular death [5]. Thus, relief of LVOTO has been the most important target for medical and surgical therapeutic efforts over the decades. Recent insight into the pathophysiology of LVOTO in HCM has led to the development of a groundbreaking approach, based on a solid pre-clinical rationale, represented by allosteric small molecule cardiac myosin inhibitors. This class of molecules represents the first disease-specific remedy in the HCM armamentarium, representing a potential game-changer in the field and allowing further understanding of the pathophysiology underlying dynamic obstruction. We here review the rationale and clinical development of Mavacamten and CK-274, discussing the potential implications for patients with obstructive HCM and beyond.

## 2. Mechanics of LVOT Obstruction

Although the LV anatomical and functional characteristics may be diverse, a recurrent scenario may be delineated. In obstructive HCM, the protruding leaflets of the mitral valve are caught by the outgoing LV flow misdirected by septal hypertrophy and, due to the consequent systolic anterior movement (SAM), make contact with the septum very early in systole, starting a vicious circle in which the creation of a gradient pushes the valve further into the outflow, increasing the gradient, and so on. Loss of mitral leaflet coaptation due to SAM leads to mitral regurgitation. Several morphological and functional abnormalities conjure to determine LVOTO including: elongated mitral valve leaflets, anterior mitral tenting by displaced, hypertrophied papillary muscles and retracted secondary chordae, reduced LVOT size, accessory LV muscle bundles, reduced LV cavity size and hyperdynamic LV [6]. Recently, the presence of a muscular mitral-aortic discontinuity has also been found to concur to LVOT obstruction in younger patients [7]. Other mechanisms include mid-ventricular obstruction and apposition of the papillary muscles directly inserted into the middle of the anterior mitral valve leaflet with the septum [8]. Whichever structural alteration subtends LVOTO, hypercontractility plays a pivotal role in generating the gradient, as demonstrated by the dropping of obstruction in HCM patients with loss of systolic performance due to disease progression. 

## 3. Available Medical Armamentarium for LVOTO

Based on this rationale, current medical therapy for symptomatic obstructive HCM is based on the use of drugs with negative inotropic and chronotropic effects. Beta blockers have been the mainstay for five decades (Class I, level of evidence, LOE, B-NR in the 2020 AHA/ACC guidelines [9]). Beta blockers are most effective for exercise-induced but less so for resting-obstruction [10,11]. Non-dihydropyridine calcium channel blockers may substitute beta blockers whenever beta blockers are ineffective or not tolerated (Verapamil, Diltiazem, Class I, LOE B-NR and C-LD respectively in the 2020 AHA/ACC guidelines [9]). However, they have a modest effect on LVOT gradient and should be avoided in patients with severe resting obstruction (>80 mmHg) associated with marked symptoms [12,13]. In the case of refractory symptoms despite beta blockers or calcium channel blocker therapy, an add-on strategy with disopyramide is recommended (Class I, LOE B-NR in the 2020 AHA/ACC guidelines [9]). 

## 4. Disopyramide

Disopyramide is a class Ia antiarrhythmic drug with negative inotropic properties, more powerful than beta blockers and verapamil in controlling LVOTO [14]. The biochemical mechanisms of benefit have been recently investigated in surgical septal samples from HCM patients [15]. Disopyramide has a sarcomere-independent negative inotropic effect mediated by multichannel inhibition (mostly of the peak Na+ current, but active also on the slow Na+ current, Ca2+ and K+ transients and ryanodine receptors). Ultimately, the drug produces lowering of systolic intracellular Ca++ levels and reduced calcium-mediated activation of the myofilaments. These effects exhibit marked antiarrhythmic activity in vitro, mediated by suppression of early and delayed afterdepolarizations and reduced transmural dispersion of repolarization [15]. Efficacy and safety of disopyramide have been shown in large multicenter registries [16,17,18], and the drug can be routinely started in the outpatient setting [19]. The most important side effects are an increase in the QTc interval, which appears to be smaller in HCM patients compared to healthy subjects [20], and anticholinergic side effects (e.g., dry mouth, constipation, urinary retention, blurred vision) that may limit patient compliance. Overall, the combination of disopyramide and beta blockers or calcium antagonists may reduce resting gradients by >50% and reduce limiting symptoms in 50% to 65% of patients, although ultimately only about 25% seem to benefit long-term [21]. In addition, even when optimized, this regime fails to address the core pathophysiology of the disease, and does not seem to affect its natural history [4]. Thus, novel advances in the medical treatment of obstructive HCM are eagerly awaited.

## 5. Targeting HCM Pathophysiology: A Novel Approach Based on Molecular Knowledge

Myosins are molecular propellers converting the chemical energy of ATP hydrolysis into the mechanical force necessary for muscle contraction [22]. In a simplified representation, each myosin dimer consists of two globular heads, and the coiled-coil tail. Myosin swings in equilibrium between two states: (1) an open-headed structure, available for actin crossbridge formation and with a high ATPase rate and (2) a folded back or super-relaxed state, not accessible for actin interaction and with low ATPase activity (Figure 1). Hence, the generated force depends on the amount of open-headed myosins [23]. Myosin missense mutations causing HCM, weaken intramolecular interactions favoring the folded state of myosin, resulting in an increase of available myosin heads and hence in a hypercontractile state. The reversible, selective allosteric inhibitor of cardiac myosin ATPase Mavacamten, on the other hand, shifts the on-off state equilibrium toward the off-folded state, causing a dose-dependent reduction in contractility [23] Mavacamten was initially studied in a murine HCM model in which it attenuated hypercontractility, and, if administered early, blunted the development of ventricular hypertrophy, fibrotic response and cardiomyocyte disarray [24]. CK-274 acts on the same principle, the main difference from Mavacamten residing in a shorter half-life.

## 6. Clinical Trials with Mavacamten

In the proof-of-concept phase 2 PIONEER-HCM trial, treatment with Mavacamten in oHCM led to improvements in post-exercise LVOT gradients, exercise capacity and symptoms, and was generally well tolerated, with most adverse effects being mild or moderate, self-limiting and judged to be unrelated to the study drug [25]. Following these promising results, the multicenter, phase 3, randomized, double blind, placebo-controlled trial EXPLORER-HCM was conducted. In this trial, 251 oHCM patients were randomized to Mavacamten or placebo on top of their previous beta-blocker or calcium channel blocker therapy (disopyramide was not allowed due to safety concerns). The primary endpoint was a composite evaluating functional capacity estimated by cardiopulmonary exercise testing and symptoms burden as estimated by the physician: patients with either a ≥1.5 mL/kg/min increase in pVO2 with ≥1 NYHA class improvement or ≥3.0 mL/kg/min increase in pVO2 with no worsening of NYHA class were considered responders. After 30 weeks of treatment, twice as many patients on Mavacamten met the primary endpoint (37% vs. 17%, *p* = 0.0005). This result was associated with marked reduction in post exercise LVOT gradient over the 30 weeks of treatment, compared to no change in the placebo group (mean post-exercise LVOT gradient difference between groups −36 mmHg, *p* < 0.0001) and improvement in functional capacity expressed by a greater increase in both pVO2 (1.4 vs. 0.6 to 2.1 mL/kg/min, *p* = 0.0006) and NYHA functional class (≥1 NYHA class improvement in 65% vs. 31% respectively in the interventional and in the placebo group, *p* < 0.0001). Furthermore, treatment with Mavacamten was associated with an amelioration in patient-reported outcomes including quality of life (Kansas City Cardiomyopathy Questionnaire, KCCQ, Score mean change from baseline 13.6 vs. 4.2, *p* < 0.0001) and HCM core symptoms, evaluated through the newly developed, disease-specific Hypertrophic Cardiomyopathy Symptom Questionnaire Shortness-of-Breath (HCMSQ-SoB) subscore (HCMSQ-SoB subscore mean change from baseline −2.8 vs. −0.9, *p* < 0.0001). Nearly three-fourths of the patients on Mavacamten showed an LVOT gradient reduction below the guideline-defined threshold for invasive therapy (post-exercise LVOT peak gradient <50 mmHg 74% vs. 21%, *p* < 0.0001), nearly 60% experienced a complete relief from obstruction (LVOT gradient <30 mmHg 57% vs. 7%, *p* < 0.0001) and about 30% reached a complete relief from both symptoms and obstruction (NYHA class I and exercise-induced gradient <30 mmHg 27.4% vs. 0.8%, *p* < 0.0001). Clinical improvement was associated with marked reduction in serum levels of N-terminal pro-brain natriuretic peptide (NTproBNP) and troponin I, two predictors of long-term outcome in HCM [26].

These results were achieved at the expense of a small average reduction in ejection fraction (mean difference −4%, 95% CI −5.5 to 2.5) and only five patients underwent protocol-driven temporary treatment discontinuation for left ventricular ejection fraction (LVEF) <50%; the latter normalized in all following wash-out, allowing study completion on a lower dose.

Mavacamten showed a favorable safety profile, overall comparable to placebo. Notably, the number of patients with severe adverse events and the absolute number of severe adverse events were actually lower in the active treatment arm compared with placebo. Two patients on Mavacamten developed stress cardiomyopathy (although one was on drug-washout), for reasons currently unknown and requiring further investigation.

Overall, EXPLORER-HCM represents a pivotal trial, in that it demonstrated for the first time the potential of a targeted molecular approach based on a solid pre-clinical rationale in HCM. Although the study was necessarily designed on surrogate end-points, due to the low rate of “hard” cardiovascular endpoints (e.g., cardiovascular mortality, heart transplantation) in this disease, its implications suggest a profound effect on several aspects of the disease, allowing hopes for long term efficacy and impact on natural history. Study limitations included the lack of information regarding disopyramide co-administration and the exclusion of severely symptomatic (NYHA class IV) patients, both areas currently under investigation (VALOR-HCM trial, A Study to Evaluate Mavacamten in Adults with Symptomatic Obstructive HCM Who Are Eligible for Septal Reduction Therapy; NCT04349072).

Very recently, in the EXPLORER-HCM cardiac magnetic resonance substudy, Mavacamten treatment was shown to induce reverse cardiac remodeling, reducing LV mass index and maximum LV wall thickness. Moreover, patients on Mavacamten presented a greater reduction in LV atrial volume index, reflecting hemodynamic improvement and a decrease in LV filling pressure. These structural changes paralleled the favorable reduction in circulating biomarkers [27].

Finally, preliminary data suggest a beneficial effect of Mavacamten also in non-obstructive HCM patients, as seen in the phase 2 MAVERICK-HCM trial. In this study, exploratory analyses showed a significative reduction in NT-proBNP, although only one-third of patients with a more severe phenotype improved in terms of NYHA class and pVO2 [28]. Main studies on selective allosteric cardiac myosin inhibitors are summarized in Table 1.

## 7. New Horizons in Selective Sarcomere Protein Modulation

Another cardiac myosin ATPase inhibitor, CK-3773274 (CK-274), is currently under investigation in the ongoing phase 2 REDWOOD-HCM trial. CK-274 has a similar mechanism of action to Mavacamten, stabilizing myosin in a weak actin-binding conformation (Figure 1), but presents a different binding site on cardiac myosin and has a shorter half-life [29]. Safety and tolerability in healthy participants have been demonstrated in a Phase 1 study that has opened the door to the Phase 2 study REDWOOD-HCM, a multicenter, randomized, placebo-controlled, double-blind, dose-finding clinical trial of CK-274 in patients with oHCM [30]. In Cohort 1, 21 patients have been randomized 2:1 to escalating doses of CK-274 (5, 10, and 15 mg once daily) or placebo. The escalation strategy included two steps at a distance of two weeks and escalation criteria included LVEF >50% and resting LVOT gradient >30 mmHg or post-Valsalva LVOT gradient >50 mmHg. The interim analysis of data from Cohort 1 of the REDWOOD-HCM trial showed that patients experienced substantial reduction in the average resting LVOT gradient as well as the post-Valsalva LVOT gradient (defined as resting gradient <30 mmHg and post-Valsalva gradient <50 mmHg) at the expense of only a modest decrease in average LVEF; no dose interruptions due to LVEF reduction below 50%, the prespecified safety threshold, were necessary. Based on these results, the trial is now assessing doses of 10 mg, 20 mg and 30 mg in Cohort 2 [31].

Targeted therapies are being developed for the broader spectrum of “sarcomeric” disease, beyond the borders of HCM and involving its “opposite” dilated cardiomyopathy. The selective cardiac myosin activator Omecamtiv mecarbil acts by stabilizing the myosin head in the open state (Figure 1), increasing the number of actin-myosin interactions and thus augmenting generated force [32]. In Phase 1 and 2 trials Omecamtiv mecarbil showed good safety and tolerability and induced an hemodynamic improvement increasing stroke volume, reducing LV volumes and NT proBNP in patients with systolic dysfunction [33,34]. The recently completed phase 3 trial Galactic-HF [35] enrolled symptomatic patients with chronic HF and EF ≤ 35%, randomized to receive oral Omecamtiv mecarbil or placebo for 48 weeks on top of standard HF therapy. A significant, although rather modest advantage was seen in the treatment arm compared to placebo, with an 8% relative risk reduction in the composite primary outcome of a heart-failure event or death from cardiovascular causes than those in the placebo group. There was no difference between groups in the secondary outcomes of cardiovascular death, all-cause death, or change in KCCQ total symptoms score [35]. Despite the limited benefit demonstrated in Galactic HF, Omecamtiv mecarbil represents an important conceptual advancement in the treatment of systolic HF and broadens the therapeutic opportunities of patients with reduced ejection fraction. Remarkably, it is the first positive inotrope drug to improve survival in cardiac patients. Future studies will hopefully identify a specific subset of patients who may gain maximum benefit from this approach.

Finally, the same molecular development pipeline is bringing hopes to patients with neuromuscular diseases, particularly with amyotrophic lateral sclerosis (ALS). Fast skeletal muscle troponin activators (FSTAs) have been shown to increase muscle force by slowing the off rate of calcium from troponin, thus increasing calcium sensitivity [36]. After the negative results of the VITALITY-ALS trial with the Tirasemtiv [37], the second generation of FSTA Reldesemtiv has shown promise [38], and is currently undergoing the Phase 3 COURAGE-ALS study [39].

## 8. Advances in the Surgical Approach to Obstructive HCM

Septal myectomy is the gold-standard strategy for the treatment of drug-refractory symptoms in oHCM patients (Class I, LOE B-NR in the 2020 AHA/ACC guidelines [9]). The intervention has been shown to improve exercise capacity [40], quality of life [41] and, most likely, long-term survival, affording mid-term survival rates comparable to the general population [42]. In expert hands, the procedure is safe, with an operative mortality of <1%, approaching zero in high-volume centers [43]. During the last two decades, important developments have taken place and contemporary surgical approaches target all components of LVOT obstruction in a tailored fashion. Specifically, the original Morrow transaortic myectomy has been modified into an extended myectomy, performed beyond septal bulge resection to the base of papillary muscles, thus redirecting the blood flow anteriorly and medially [44]. An extended septal myectomy gains exposure of the papillary muscles that can be thinned and mobilized from the free wall of the LV, excising abnormal muscular connections [6,8]. Recently, and of particular relevance for patients with LVOT obstruction associated with minimal LV hypertrophy, in whom the myectomy is necessarily shallow, the “chordal cutting” technique has been introduced by Ferrazzi et al. [45]. This approach consists of appropriate cutting of retracted and fibrotic secondary chordae tenting up the anterior mitral leaflet into the outflow tract, leading to satisfactory relief of obstruction without the need for extensive removal of septal myocardium [8,45]. Technical variants introduced by other groups have focused their attention on the abnormalities of the mitral leaflets. The elongated anterior mitral leaflet with excess of tissue in the horizontal axis may be plicated, placing three to four sutures in a horizontal orientation through the fibrotic area corresponding to the contact point with the ventricular septum, thus reducing leaflet length and leaflet/chordal slack [44]. Very long residual leaflets that extend beyond the coaptation point of the mitral valve and protrude into the left ventricle can be removed with their slack and thinned out chordae through a residual leaflet excision termed “ReLex”, introduced by Swistel et al., leaving about 1 cm of coaptation length to maintain valvular competence [46]. At our institution, when the posterior mitral leaflet significantly exceeds the anterior in length, a triangular or quadrangular re-section and folding/sliding plasty is performed. In the case of annular calcification or leaflet fibrosis, manipulating the tissue is not feasible and mitral valve replacement may be necessary [8]. The predominant approach for septal myectomy remains a full median sternotomy. Nonetheless several centers, including ours, opt for a minimally invasive technique using a ministernotomy [47]. The latter option is still not feasible in case of concomitant surgery requiring a wider exposure such as mitral valve repair/replacement, coronary artery bypass grafting, or the Cox-Maze procedure for atrial fibrillation (AF). The addition of the surgical Cox-Maze IV procedure has shown particularly good results for control of AF in HCM patients, with 65% of treated patients free from symptomatic AF recurrence at five years of follow up [48].

The timing of intervention is an important and still unresolved issue. We have recently shown that a prolonged exposure to LVOTO is detrimental, even in the absence of severe symptoms. Specifically, a delay of more than five years from first gradient detection to septal reduction therapy (SRT) was associated with doubling of the risk of cardiovascular mortality and disease progression [49]. In a recent retrospective study from Alashi et al. [50], mildly symptomatic patients with drug intolerance or impaired exercise capacity on stress echocardiography referred for surgery showed improved outcome compared to those following the conventional indication (i.e., later referral) [50]. However, the procedures were associated with a not negligible risk of severe adverse events in an overall slightly symptomatic population, and only a small number of patients were on optimal medical therapy (only 4% on disopyramide). Therefore, caution is warranted before these results are generalized to the entire oHCM population, particularly outside high-volume centers [21]. HF, other than LVOTO, may be caused by progressive LV dysfunction or restrictive evolution, with a severely unfavorable prognosis in both cases. This scenario, although infrequent, tends to respond poorly to conventional HF treatment and heart transplant becomes the only option. Active surveillance with imaging techniques, cardiopulmonary testing and biomarkers is needed to early identify the candidates for advanced treatment and reduce the probability of heart transplant exclusion or failure. Transplant referral does not absolutely require reduced EF, in fact patients with restrictive physiology may develop severe HF despite LVEF values well above 50%, furthermore, due to the hyperdynamic nature of the disease, LVEF values approaching 55% should already be regarded as a wakeup call for end-stage evolution [9,51].

## 9. Modifiable Cardiovascular Risk Factors

The cardiovascular risk factors control in patients with HCM should be strict in order to avoid the development of a double, ischemic and inherited, cardiomyopathy [52]. In fact, the increased myocardial mass, arteriolar remodeling and microvascular dysfunction associated with higher oxygen demand make these patients particularly vulnerable to ischemic insults and, unsurprisingly, coronary artery disease is associated with a worse prognosis [53]. Among the risk factors, attention should be paid to obesity. The latter is highly prevalent in HCM patients and is associated with higher inducible gradients, more rapid clinical progression, and worsening of HF symptoms [54,55]. Hence, obesity treatment, including tailored exercise programs, nutritional counselling, pharmacological therapy and bariatric surgery may be needed to prevent disease related complications.

## 10. A Treatment Algorithm for the Next Decade

The stepped management proposed by the recent AHA/ACC guidelines defines the best of possible worlds in HCM, balancing risks and benefit in a long-term perspective [9]. In the light of new therapeutic approaches, a hypothetical prediction of how the algorithm may change, following the clinical introduction of Mavacamten (pending FDA approval) and CK-274 (still under investigation) is depicted in Figure 2. In patients with obstructive HCM, first line management with beta blockers or calcium channel blockers and, in case of unsatisfactory effect, disopyramide, is not expected to change. If the combination therapy results inadequate, a switch from disopyramide to a myosin inhibitor might be considered. SRTs remain indicated in the presence of drug-refractory symptoms—hopefully less frequent with the introduction of the new agents. Surgical myectomy should be preferred over alcohol septal ablation (ASA) when local expertise is available. Atrioventricular (AV) sequential pacing may be considered in selected (generally older) patients, while the MitraClip and other percutaneous approaches have been attempted but should be considered investigational.

ASA is the first option in patients not eligible for surgery (Class I, LOE C-LD in the 2020 AHA/ACC guidelines [9]). This approach implies the delivery of absolute ethanol to the first (and occasionally second) septal perforator branch of the left anterior descending coronary artery inducing a localized myocardial infarction in the hypertrophied LV septum [56]. ASA and myectomy share similarly low long-term mortality and (aborted) sudden cardiac death rates, despite the initial safety concerns regarding the creation of a potentially arrhythmogenic ablation scar [57]. ASA implies an increased risk of pacemaker implantation, greater post-procedural LVOT gradients and thus higher rates of reintervention compared to surgery [57,58] In order to avoid alcohol dispersion into adjacent vessels through collaterals, coil embolization of one or more septal arteries has been proposed as an alternative to ASA. This technique has been proven to be safe and durable over the long term in a small oHCM population [59], but further large-scale studies are needed to confirm these findings.

AV sequential pacing was considered an obsolete technique until long term studies reported a sustained effect on LVOT gradient reduction [60,61]. In the most recent study by Roman et al. [62], AV sequential pacing was associated with an immediate peak LVOT gradient reduction of 50% and 60% after 8 years of follow up, coupled with a sustained improvement of at least 2 NYHA classes in nearly 80% of patients. Therefore, this option should be considered in high surgical risk patients who already have a dual chamber pacemaker.

In patients with refractory HF with reduced ejection fraction and chronic severe secondary mitral regurgitation, edge to edge mitral valve repair with the use of the MitraClip system (Abbott Vascular, Santa Clara, CA, USA) is a reasonable approach (COR 2a, LOE B-R in the ACC/AHA guidelines [63]). This technique has been translated in the oHCM treatment; in fact, the percutaneous plication of the mitral valve leaflets may prevent mitral-septal contact and relieve mitral regurgitation. In a series of six elderly patients with oHCM from Sorajja et al. [64] the mitral clip implantation significantly reduced LVOT gradients, systolic anterior motion-related mitral valve regurgitation and relieved symptoms. Nonetheless, half of patients presented markedly elevated systolic velocities across the LVOT on follow up echocardiography, probably due to Doppler gradient overestimation related to pressure recovery phenomena, and two persistently symptomatic patients developed post procedural moderate mitral stenosis. oHCM patients present a two-fold relative risk of sudden cardiac death than non-obstructive patients [5,65]. However, the absolute risk remains very small and the LVOT gradient may not be considered the sole determinant of decisions to implant cardioverter–defibrillators prophylactically. Indeed, LVOT gradient has been included as a continuous variable in the European risk prediction model for sudden cardiac death at five years, a useful support for implantable cardioverter-defibrillator decision-making [66].

## 11. Conclusions

LVOT obstruction is the most prevalent cause of heart failure in patients with HCM, and its relief is a fundamental therapeutic aim requiring a concerted effort by physicians and surgeons, fulfilling the “HCM Heart Team” concept [67]. Ground-breaking advances in our understanding of HCM have led to the development of a revolutionary pharmacological approach involving allosteric myosin inhibitors. The pivotal results of EXPLORER-HCM justify cautious optimism for the future of HCM patients, despite the need for further evidence regarding long-term efficacy and safety of these agents. Whatever the outcome, a new and much awaited era has begun for our patients, likely to see the irresistible emergence of precision medicine in cardiomyopathies.

## 12. What’s New in This Review?

Hypertrophic cardiomyopathy (HCM) is a genetic heart disease characterized by various degrees of left ventricular hypertrophy, hyperdynamic contractility and mitral valve abnormalities that may lead to left ventricular outflow tract obstruction (LVOTO).

LVOTO is the main cause of disabling symptoms and heart failure in HCM patients and its relief is a fundamental therapeutic aim requiring a concerted effort by physicians and surgeons.

From a solid pre-clinical rationale, a targeted therapy for LVOTO represented by allosteric cardiac myosin inhibitors has been developed. The new drugs have been studied in recent clinical trials that showed an improvement in functional capacity, a reduction in LVOT gradient and related symptoms evaluated by both physicians and patients.

The progress in medical therapy parallels the advance towards a tailored surgical approach that remains the gold-standard for the treatment of drug-refractory symptoms in obstructive HCM patients.

The new targeted medical therapies in HCM might have set the basis for the emergence of precision medicine in cardiomyopathies.

## Figures and Tables

**Figure 1 jcm-10-00951-f001:**
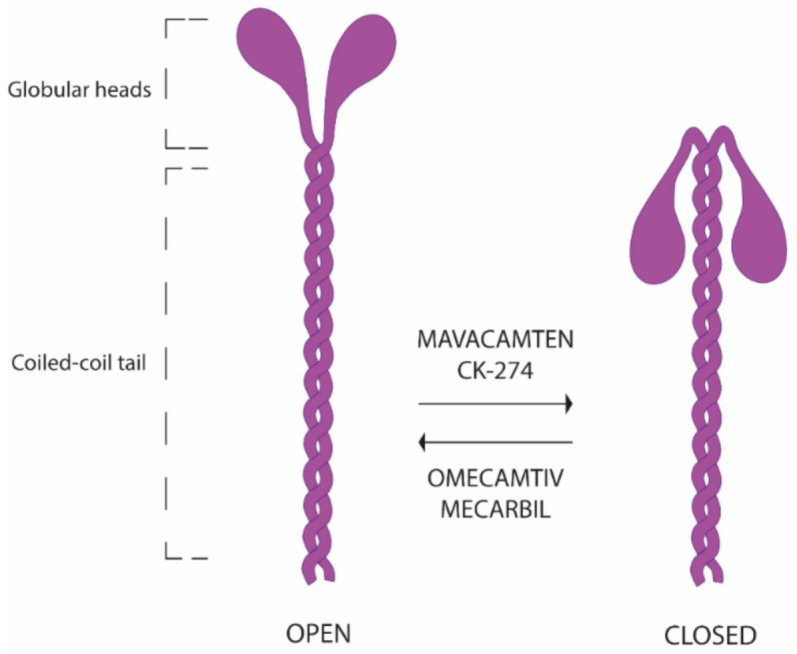
Structural model of the open on-state and the closed-off state of human beta cardiac myosin. Mavacamten and CK-274 shift the on-off state equilibrium toward the off-folded state, ultimately reducing contractility, in the opposite way Omecamtiv mecarbil stabilizes the open state increasing contractility.

**Figure 2 jcm-10-00951-f002:**
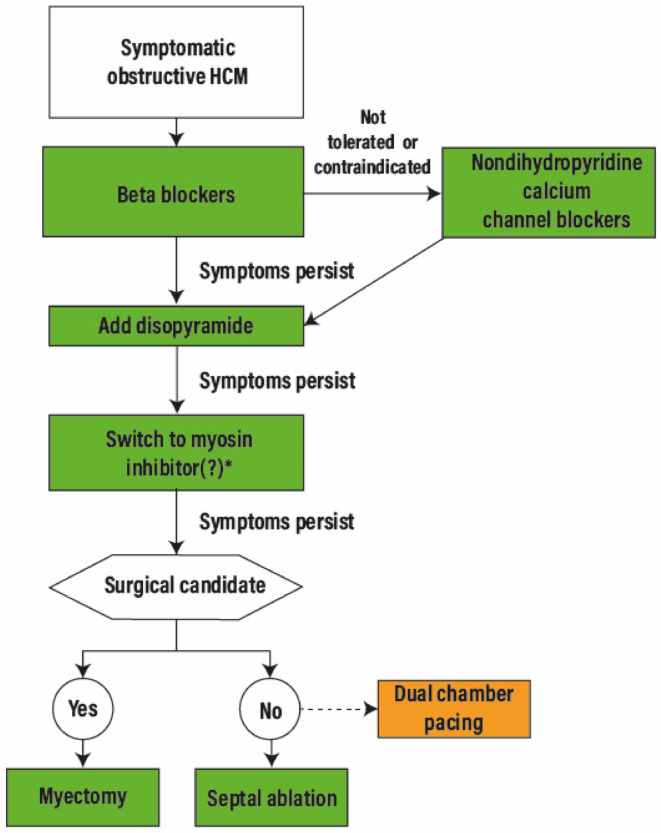
Possible treatment algorithm for obstructive Hypertrophic Cardiomyopathy therapy. HCM: Hypertrophic Cardiomyopathy. * Mavacamten has been submitted for Food and Drugs administration approval. See the text for the flow chart description.

**Table 1 jcm-10-00951-t001:** Main studies on selective allosteric cardiac myosin inhibitors. Abbreviations: CI: confidence interval, HCM: hypertrophic cardiomyopathy, HCMSQ-SoB: HCM Symptom Questionnaire Shortness-of-Breath subscore, KCCQ-CSS: Kansas City Cardiomyopathy Questionnaire, LVEF: left ventricular ejection fraction, LVOT: left ventricular outflow tract, MG: Mavacamten group, *n*: score, NTproBNP: N-Terminal pro-Brain Natriuretic Peptide, NYHA: New York Heart Association functional class, oHCM: obstructive Hypertrophic Cardiomyopathy, PG: placebo group, pt: patients, SRT: septal reduction therapy, y: years.

Study Name and Status	Molecule	Study	Population	Primary Endpoints	Secondary Endpoints
PIONEER-HCM Status: completed	Mavacamten MyoKardia, Inc.	Multi-center phase II open-label non-randomized 2 sequential cohorts (A and B) 12-weeks treatment phase followed by a 4-weeks post-treatment phase	21 oHCM patients Cohort A: 56y Cohort B: 58y 57% men 57% NYHA II 43% NYHA III	Cohort A: mean post-exercise LVOT gradient decreased from 103 ± 50 mmHg to 19 ± 13 mmHg at 12 weeks (*p* = 0.008) Cohort B: mean LVOT gradient decreased from 86 ± 43 mmHg to 64 ± 26 mmHg (*p* = 0.020)	Cohort A: Resting LVEF reduction −15% (CI, −23% to −6%). Peak VO2 increased by a mean of 3.5 mL/kg/min (CI, 1.2 to 5.9 mL/kg/min). Cohort B: mean change in resting LVEF −6% (CI, −10% to −1%). Peak VO2 increased by a mean of 1.7 mL/kg/min (CI, 0.03 to 3.3 mL/kg/min). Dyspnea scores improved in both cohorts.
EXPLORER-HCM Status: completed	Mavacamten MyoKardia, Inc.	Multi-center phase III randomised double-blind placebo-controlled	251 oHCM, Mean age 58 ± 11 years, 59% men	≥1.5 mL/kg/min increase in pVO2 with ≥1 NYHA class improvement OR ≥ 3.0 mL/kg/min increase in pVO2 with no worsening of NYHA class in 37% of pt in MG vs. 17% in PG (*p* <0.0005)	Post-exercise LVOT gradient −47 mmHg in MG vs. −10 mmHg in PG (*p* < 0.0001) pVO_2_ +1.4 mL/kg/min in MG vs. −0.05 mL/kg/min in PG (*p* < 0.0006) ≥ NYHA class improvement 65% pt in MG vs. 31% pt in PG (*p* < 0.0001) KCCQ-CSS (*n*): +14 in MG vs. +4 in PG (*p* < 0.0001) HCMSQ-SoB (*n*) (negative better): −2.8 in MG vs. −0.9 in PG (*p* < 0.0001)
MAVERICK-HCM Status: completed	Mavacamten MyoKardia, Inc.	Multi-center phase II randomized double-blind placebo-controlled	59 non-oHCM, mean age 54 ± 14 y, 58% women	Serious adverse events occurred in 10% pt in MG and in 21% of pt in PG 5 pt in MG had reversible reduction in LVEF ≤45%.	NTproBNP decreased by 53% in the MG vs. 1% in the PG, (*p* = 0.0005). Troponin I decreased by 34% in the MG vs. a 4% increase in the PG, (*p* = 0.009).
VALOR-HCM Status: on-going	Mavacamten MyoKardia, Inc.	Multi-center phase III randomized double-blind placebo-controlled	oHCM	Number of subjects who proceed or remain guideline eligible for SRT within week 16	Number of subjects who proceed or remain guideline eligible for SRT within week 32; change from baseline to week 16 in NYHA, KCCQ-CSS, NTproBNP, troponin, LVOT gradient
MAVA-LTE Status: ongoing	Mavacamten MyoKardia, Inc.	Multi-center phase III randomized	A Long-Term Safety Extension Study of Mavacamten in who have completed the MAVERICK-HCM or EXPLORER-HCM Trials	Frequency and severity of treatment-emergent adverse events and serious adverse events	
REDWOOD-HCM Status: ongoing	CK-274 Cytokinetics Inc.	Multi-center phase II randomized double-blind placebo-controlled	oHCM	Safety and tolerability	Concentration-response and dose-response on the resting and post-Valsalva LVOT gradient; effect on NTproBNP and NYHA.

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
