# Peer review of "Emerging Medical Treatment for Hypertrophic Cardiomyopathy"

_jcm, 2021, doi:10.3390/jcm10050951_

Round 1

Reviewer 1 Report

The manuscript Argiro A et al is a good comprehensive review on current treatment available for patients suffering from LVOT due to hypertrophic cardiomyopathy. The review is well written and focussed. I have following concerns for the manuscript:

  1. The clinical trials of mavacmaten and CK-274 has been described well but requires the outcomes in a good table format, which will be easy to understand.
  2. The authors have discussed only about mutation based HCM treatment. Discuss about pathophysiological HCM by obesity or atherosclerosis and show light on their treatment as well.
  3. Please provide key highlights of the review like what new information this manuscript is providing. 

Reviewer 2 Report

This manuscript provides a useful review of treatments for hypertrophic cardiomyopathies with some background on the mechanism of the pathology. There are a few places where the manuscript would benefit from corrections and treatments that should be mentioned at least briefly in comparison to those that are covered in more detail.

  1. Figure 1 illustrates the OPEN (also known as ON) and CLOSED (also known as OFF) states of the myosin molecule, but on lines 218-219, the text incorrectly refers to the former as the “pre-powerstroke” state: “The selective cardiac myosin activator Omecamtiv mecarbil acts by stabilizing the myosin head in the “pre-powerstroke” state (Figure 1)”. The pre-powerstroke state is a designation of a specific myosin conformation of the OPEN state in the part of the ATPase just prior to phosphate release. A more suitable term such as that used in Figure 1 should replace “pre-powerstroke” here.
  2. The manuscript correctly points out the diversity in phenotypes associated with different hypertrophic cardiomyopathy cases such as “morphological patterns and severity are extremely different” and “The clinical course of HCM is characteristically diverse, ranging from completely asymptomatic subjects to advanced heart failure due to severe left ventricular outflow tract (LVOT) obstruction, restrictive physiology or left ventricular systolic dysfunction.” In the very next section on Mechanics of LVOT obstruction, however, the reader is left with the impression the course of the hypertrophic cardiomyopathy always follows the same mechanical sequence of events. Although the mechanical description is useful and quite plausible for some patients, the scenario requires some reservations with regards to its universal applicability so as to reduce the clash with the previous section.
  3. There should be some mention of cardiac transplantations as a treatment sometimes used for advanced end-stage disease, since it is often reported in the literature for severe cases.
  4. The only device implant mentioned is the pacemaker; however, implantable cardioverter defibrillators and cardiac resynchronization therapy devices deserve a brief mention too.
